# Impact of CPAP Therapy on New Inflammation Biomarkers

**DOI:** 10.3390/jcm11206113

**Published:** 2022-10-17

**Authors:** Tea Friščić, Marko Perčić, Domagoj Vidović, Andrija Štajduhar, Edvard Galić

**Affiliations:** 1Clinical Hospital Sveti Duh, 10000 Zagreb, Croatia; 2University Psychiatric Hospital Vrapče, 10000 Zagreb, Croatia; 3School of Medicine, University of Zagreb, 10000 Zagreb, Croatia

**Keywords:** obstructive sleep apnea, continuous positive airway pressure, inflammation, cardiovascular risk, atherogenesis

## Abstract

Obstructive sleep apnea (OSA) is a common sleep disorder leading to increased risk of developing cardiovascular diseases (CVDs) by supporting a low-grade chronic inflammation as one of the pathological mechanisms. The continuous positive airway pressure (CPAP) device is used as an effective treatment for moderate and severe OSA. Neutrophil–lymphocyte ratio (NLR), platelet–lymphocyte ratio (PLR), white blood cell–mean platelet volume ratio (WMR), C-reactive protein–albumin ratio (CAR) and fibrinogen–albumin ratio (FAR) are new potential inflammatory biomarkers that are widely available and were shown to be possibly favorable screening or follow-up tools for moderate- or severe-grade OSA, as well as indirect indicators for cardiovascular risk. Our study evaluated the impact of CPAP therapy in patients with severe OSA and acceptable therapy adherence on NLR, PLR, WMR, FAR and CAR. Of 57 patients who were initially enrolled and had no exclusion criteria, 37 had a satisfactory CPAP adherence (usage of ≥4 h per night) after a minimum of 6 months of therapy. There was a statistically significant difference in NLR (2.122 ± 0.745 before therapy vs. 1.888 ± 0.735 after therapy) and FAR (86.445 ± 18.763 before therapy vs. 77.321 ± 19.133 after therapy) suggesting a positive effect of the CPAP therapy on chronic inflammatory states, thereby possibly reducing cardiovascular risk.

## 1. Introduction

Obstructive sleep apnea (OSA) is a common sleep disorder with an estimated average prevalence of 22% (9–37%) in men and 17% (4–50%) in women [1,2]. It is caused by a partial or complete upper airway collapse during sleep, which leads to intrathoracic pressure swings, intermittent hypoxia and sleep fragmentation with pathophysiological consequences including increased oxidative stress, adrenergic activation, endothelial dysfunction, hypercoagulability, metabolic dysregulation, endocrine activation and chronic systemic inflammation [3]. As a result, OSA leads to an increased risk of developing cardiovascular diseases (CVDs) including heart failure (HF), myocardial infarction (MI), atrial fibrillation (AF), stroke, diabetes, chronic kidney disease (CKD), neoplasms, polycystic ovary syndrome and erectile dysfunction. OSA can have a great impact on the quality of life (QoL), not only affecting physical health and cognitive function but also affecting social and occupational spheres [4]. The diagnosis and severity of OSA is based on the apnea–hypopnea index (AHI), which is calculated from the data of a polysomnography study (PSG) or a home sleep apnea test (HSAT). Mild OSA is characterized as an AHI from 5 to 14 (events per hour), moderate by an AHI from 15 to 29, and severe OSA by an AHI >30 [5]. The continuous positive airway pressure (CPAP) device is used as an effective treatment for moderate and severe OSA leading to an improvement in daytime sleepiness and reducing AHI, but the beneficial effects on the cardiovascular system are still not clear [6].

Low-grade chronic systemic inflammation in OSA patients supports the progression of atherogenesis and further development of CVDs. Several studies showed that, in OSA patients, the levels of circulating markers of inflammation, such as C-reactive protein (CRP), cytokines (interleukin 6 (IL-6), interleukin 8 (IL-8), tumor necrosis factor α (TNF-α)) and adhesion molecules are elevated, but the impact of CPAP therapy is still controversial [7,8].

New potential biomarkers of inflammation have emerged, being a theoretically valuable screening tool for moderate- or severe-grade OSA or as a follow-up tool for therapy adherence and efficiency, as well as indirect indicators for cardiovascular risk. They are ratios calculated from the complete blood count (neutrophil–lymphocyte ratio (NLR), platelet–lymphocyte ratio (PLR), white blood cell–mean platelet volume ratio (WMR)), or basic biochemistry panel (CRP–albumin ratio (CAR), fibrinogen–albumin ratio (FAR)). The dynamics of those biomarkers related to the grade of OSA and the impact of CPAP therapy still has to be investigated because the results from studies published thus far are insufficient [9,10].

## 2. Materials and Methods

### 2.1. Patient Selection

This prospective, controlled, clinical study recruited patients from a clinic for sleep disorders during the period between July 2019 and November 2020, after an overnight hospital PSG (Respironics Alice^®^ 6: Sleepware G3 3.7.3, Murrysville, PA, USA). Results were interpreted according to American Academy of Sleep Medicine rules for scoring respiratory events. Apnea in adults was scored when there was a drop in the peak signal excursion by ≥90% of pre-event baseline for ≥10 s. Hypopnea was scored when the peak signal excursions drop by ≥30% of pre-event baseline for ≥10 s in association with either ≥3% arterial oxygen desaturation or an arousal [11]. All patients who had AHI ≥ 30 and no exclusion criteria (younger than 18 years old, pregnancy, severe chronic obstructive pulmonary disease (COPD), severe CKD, AF, acute coronary syndrome (ACS)/stroke/transient ischemic attack (TIA) in the last 6 months, anticoagulation/antiplatelet therapy, psychiatric causes of sleep disorder, central sleep apnea, systemic, autoimmune, inflammatory and/or nutritional disease that might affect the level of blood parameters and a history of malignancy) were enrolled (Figure 1). From the 58 who initially entered, 57 were enrolled, i.e., did not have any of the exclusion criteria after the initial cardiovascular assessment (one had newly diagnosed AF). All patients signed a written consent after they had been informed of the nature and the purpose of the study, as well as the potential risks for them as patients. The study was approved by the local ethics committee. The initial assessment and data collection were done immediately after PSG including anthropometric measures, Epworth sleepiness scale (ESS), medical history and routine blood tests. Patient follow up was scheduled after a minimum of 6 months of CPAP device usage. The exclusion criterion for CPAP adherence was the usage of the device for less than an average of 4 h per night (data obtained from the device’s memory card). Of 57 patients, 37 surpassed the exclusion criteria, and their data were evaluated in this study.

### 2.2. Laboratory Methods

The blood samples were taken in the morning between 7 and 10 a.m., after 12 h of fasting. They were analyzed within 30 min after sampling. An automated blood cell counter was used to determine the complete blood count parameter levels (Advia 2120i (Siemens Healthcare, Erlangen, Germany)). Serum albumin levels were analyzed with automatic photometry commercial kits (AU680 (Beckman Coulter Inc., Brea, CA, USA)), CRP levels with a latex immunoturbidimetric assay (AU680 (Beckman Coulter Inc., Brea, CA, USA)), fibrinogen levels with the modified Clauss method (BCS XP (Siemens Healthcare, Erlangen, Germany)), lipoprotein lipid levels with the enzyme method (AU680 (Beckman Coulter Inc., Brea, CA, USA)) and N-terminal prohormone of brain natriuretic peptide (NTproBNP) levels with an electrochemiluminescence immunoassay (Cobas e411 (Roche Diagnostics, Mannheim, Germany)). Values of NLR, PLR, WMR, CAR and FAR were calculated as ratios from the obtained data.

### 2.3. Statistical Analysis

In our study, we had 37 patients who had good compliance to the CPAP therapy, which is a sample size that yields high power for NLR and FAR variables (post hoc achieved power of 1.0) due to the large effect size (d = 2.4). For PLR and CAR, the achieved power was 0.4. The normality of the data distribution was tested using the Shapiro–Wilk test. Continuous variables with normal distributions were expressed as means and standard deviation, while the ones with non-normal distributions were summarized as medians and interquartile range. Categorical variables were expressed as numbers (percentages). Differences in variables were tested using paired-samples t-test or Wilcoxon test, depending on the data normality. For the evaluation of relationships between the variables, Pearson’s correlation coefficient and Mann–Whitney U-test were used. Differences were considered statistically significant at *p* < 0.05. Statistical analysis was performed by Python version 3.8 programming language using open-source libraries.

## 3. Results

Our study group consisted of 37 patients (78% (29) men) who had a satisfactory adherence to CPAP therapy (using it on average ≥4 h per night). The characteristics of the study group are summarized in Table 1.

All patients were diagnosed with severe OSA with an AHI of 58.4 ± 22. The ESS score was 10.6 ± 5.2. The mean age was 53 ± 10 years. The majority of the patients were obese (average body mass index (BMI) 34.4 ± 6.1, 76% (28) obese), and 27% (10) of them were smokers. Hypertension was the most frequent amongst the comorbidities (51%), followed by diabetes (16%). AHI was mainly the result of obstructive (not central) apneas and hypopneas. The average time of CPAP therapy during the night was (322.3 ± 51.3 min) and the AHI after therapy was 4 (2.9–5).

The clinical laboratory values of the study group, including complete blood count variables, NLR, PLR, WMR, CAR, FAR, lipid profile, fibrinogen, albumin and polysomnographic variables, before and after a minimum of 6 months of using CPAP are shown in Table 2.

There was a statically significant difference in values before and after CPAP therapy in RDW (12.9 (12.5–13.6) vs. 13.3 (12.9–13.7)), relative number of lymphocytes (0.305 ± 0.067 vs. 0.331 ± 0.084), relative and absolute number of neutrophils (0.6 ± 0.073 vs. 0.569 ± 0.09 and 4.07 (3.14–5.83) vs. 3.83 (2.82–5.11), respectively), NLR (2.122 ± 0.745 vs. 1.888 ± 0.735), albumin (42.595 ± 2.713 vs. 43.811 ± 3.017), fibrinogen (3.665 ± 0.752 vs. 3.365 ± 0.771), FAR (86.445 ± 18.763 vs. 77.321 ± 19.133), NTproBNP (45.27 (28.33–93.81) vs. 35 (30.99–59.88)), total cholesterol (5.408 ± 0.98 vs. 5.052 ± 0.87) and low-density lipoprotein (LDL) (3.52 (3.18–4.28) vs. 3.59 (2.92–3.77)). There was no statistically significant difference in mean platelet volume (MPV), PLR, WMR, CRP, CAR and BMI. Five patients were taking statin therapy before enrollment; one of them had the dosage up-titrated, and two of them started statin therapy after the initial recruitment. The results showed a statistically significant correlation between the initial (before CPAP) values of NLR and CRP, fibrinogen and FAR (r = 0.4093, *p* = 0.0119; r = 0.4678, *p* = 0.0035 and r = 0.5466, *p* = 0.0005, respectively), as seen in Figure 2. In addition, there was also a significant correlation between FAR and AHI, BMI and CRP (r = 0.3617, *p* = 0.0278; r = 0.4161, *p* = 0.0104 and r = 0.6198, *p* = 0.0000, respectively), as seen in Figure 3. In patients with hypertension, NLR and FAR had significantly higher values (MW = 112.0000, *p* = 0.0377 and MW = 112.0000, *p* = 0.0377, respectively) (Figure 4). Interestingly, FAR was higher in women (MW = 68.0000, *p* = 0.0398) and NLR in men (MW = 68.0000, *p* = 0.0398) (Figure 5). There was a significant correlation between the difference in PLR and the difference in NLR before and after therapy (r = 0.4305, *p* = 0.0078), the difference in FAR and the difference in CRP before and after therapy (r = 0.4118, *p* = 0.0113) and difference in FAR before and after therapy and AHI (r = −0.3741, *p* = 0.0226), confirming that the decline in new inflammatory markers follows the decline in already established inflammatory markers. Correlation between FAR and AHI shows a potentially greater benefit of the therapy in lowering inflammation in patients with more severe disease. This is an exploratory study; therefore, multiplicity corrections were not done.

## 4. Discussion

Chronic systemic inflammation of a low grade is one of the mechanisms, together with increased oxidative stress, dysregulation of autonomic nervous system and metabolic alterations, involved in pathogenesis of OSA resulting in advancing atherogenesis, increased risk for CVDs such as hypertension, stroke and coronary artery disease (CAD). Moreover, OSA is associated with cardiovascular risk factors such as dyslipidemia, insulin-resistance and metabolic syndrome, all being conditions where inflammation has a crucial role [12]. Some studies showed that OSA is a risk factor for cognitive decline in older patients including mild cognitive impairment (MCI), vascular dementia and Alzheimer’s disease (AD). There are also some signals indicating that CPAP therapy may improve cognitive function and slow down the progression of MCI and AD [13,14]. CPAP is a widely used, effective treatment strategy for relieving dyspnea. Its positive effects on clinical outcomes seem logical considering the decline in number of intermittent apneas and hypopneas—the primary pathological event. A study by Baessler et al. showed decreased levels of inflammatory markers such as CRP, IL-6 and TNF-α after using CPAP therapy [15]. Everyday clinical practice usually does not involve the evaluation of some of the markers mentioned, but new inflammation markers correlating with cardiovascular risk have emerged that can quickly be calculated from the complete blood count data.

The NLR is calculated by dividing the neutrophil count by the lymphocyte count. Neutrophils are involved in the pathogenesis of atherosclerosis by releasing granule proteins, recruiting monocytes into atherosclerotic lesions and promoting foam cell formation. A decreased lymphocyte count is a result of cortisol release due to the activated sympathetic nervous system response during systemic stress [16]. According to several studies, the NLR was an independent predictor for major adverse cardiovascular events (MACEs) and mortality in patients with ACS, after percutaneous coronary intervention (PCI) and coronary artery bypass graft (CABG) [17]. Furthermore, Park et al. showed that higher NLR was independently associated with arterial stiffness and coronary calcium score (CCS) [18], providing a signal that NLR is not only useful as a marker that in acute states but also as a marker for chronic states, i.e., atherogenesis. Some authors suggest that the NLR values decrease significantly in patients using CPAP regardless of adherence but with a more direct relationship in those who use it at least 4 h a day [19]. On the contrary, a meta-analysis of two studies comparing NLR before and after CPAP treatment found no significant difference in NLR values [20]. Our results showed a statistically significant decline in NLR after CPAP therapy and also a decline in relative number of neutrophils (numerator) and an increase in the relative lymphocyte number (denominator), thereby suggesting that the therapy reduced the level of chronic inflammation, which could lead to a lowering of the risk for CVD. Significant correlation of NLR and well-established inflammatory markers such as fibrinogen and CRP suggest that NLR could be a valid inflammatory marker. When considering cardiovascular risk factors, NLR was higher in men and patients with hypertension.

The PLR, a ratio between the platelet and lymphocyte number, was studied as a new marker of inflammation by combining the increase in platelet counts via their promotion, production and activation by inflammatory cytokines and decrease in lymphocyte counts reflecting the inflammatory pathway. Activated platelets secrete chemokines and cytokines, thus acting similar to inflammatory cells. They not only have a crucial role in acute plaque rupture and subsequent local thrombus formation but also in atherogenesis itself [21]. Several studies showed that PLR is a good predictor of MACE, prognosis and mortality in ACS treated with PCI or CABG, but also correlates with the severity of stable CAD and is higher in HF patients [22]. Koseoglu et al. conducted a cohort study on 424 patients and reported that PLR was significantly higher in patients with OSA and CVD compared with those without CVD and is strongly associated with the severity of OSA [23]. Our study showed a decline in PLR and WMR in patients after CPAP therapy, although without reaching statistical significance. Moreover, we did not find a significant difference in MPV, which is a measure of platelet size and activity.

Fibrinogen is a serum glycoprotein that has a major role in the inflammation pathway and coagulation cascade. It is an acute-phase protein synthesized by the liver with rising levels in settings of acute disease, such as bacterial infections and trauma. Fibrinogen levels are elevated not only in acute settings but also in chronic low-grade inflammation states. Fibrinogen modulates the inflammatory process by inducing synthesis of the proinflammatory cytokines IL-6 and TNF-α from peripheral blood mononuclear cells, thereby supporting vascular inflammation, oxidation, subendothelial aggregation of LDL and migration of vascular smooth muscle cells, which are crucial steps in atherogenesis [24,25,26]. Elevated fibrinogen levels can lead to an increased risk of thrombosis due to increased blood viscosity resulting in slow blood flow and aggregation of erythrocytes and platelets [27]. Furthermore, higher levels of fibrinogen are associated with the risk of CVD, MACE and presence and severity of CAD [28,29,30]. Serum albumin has several important biological functions. It is a carrier of many endogenous and exogenous substances, and it contributes to fluid balance, being one of the main components of plasma colloid osmotic pressure. It also exerts anticoagulant and antiplatelet aggregation activity and is a very important antioxidant and anti-inflammatory factor. Hypoalbuminemia is a well-established independent prognostic factor for CVD in terms of risk, prognosis and adverse outcomes [31].

FAR was proposed as a new marker of inflammation because it reflects both hyperfibrinogenemia and hypoalbuminemia. Recent studies have indicated that FAR is associated with the severity of coronary stenosis, in-stent restenosis and poorly developed coronary collateral circulation in patients with stable CAD and is an independent predictor of all-cause mortality in ACS [25,32,33,34]. A study by Hizli et al. showed that higher FAR and CAR values may be predictive markers for inflammation in patients with moderate to severe OSA, also predicting its severity [9]. Our results showed a significant correlation between FAR and AHI, i.e., severity of OSA, BMI, CRP and hypertension. This finding is suggestive of FAR being a good marker of inflammation, especially in hypertensive, obese patients with high AHI. In addition, we found a statistically significant decline in FAR and also in fibrinogen (numerator) and an increase in albumin (denominator) before and after CPAP therapy. The values of CAR also declined but did not reach statistical significance.

Recently published secondary analysis of the RICCADSA (Randomized Intervention with CPAP in Coronary Artery Disease and obstructive Sleep Apnea) trial showed that CPAP had no lipid-lowering effect in revascularized CAD patients with non-sleepy OSA [35]. An observational study using data from the European Sleep Apnea Database demonstrated a reduction of total cholesterol levels after long-term CPAP treatment [36]. Our study showed a statistically significant decline in total cholesterol after CPAP therapy. Triglyceride levels declined but did not reach statistical significance. LDL levels showed a small but statistically significant increase. In addition, some of our patients were taking statin therapy with dosage up-titration before the follow up. Thus, a lipid-lowering effect of CPAP therapy could not be determined.

Results of a meta-analysis that included 18 randomized clinical trials (RCTs) with more than 4000 OSA patients on CPAP therapy did not show a significant decrease of cardiovascular event risk compared with the control group but only a nonsignificant trend of lower rate of death and stroke [37]. Similarly, a newer meta-analysis of 8 RCTs and 5817 patients showed no evidence that CPAP therapy improves CV outcomes [38]. Another meta-analysis reported a reduced risk of MACE in a subgroup that used a CPAP device for more than 4 h per night (RR 0.70, 95% CI 0.52 to 0.94, *p* = 0.02), suggesting that the duration of CPAP therapy could be essential for improving CV outcomes [39].

This study has several limitations. Foremost is the small sample number, which is a relatively common limitation in OSA studies, especially when CPAP therapy is involved. Secondly, we did not measure high-sensitive CRP levels, which might have influenced the results. Lastly, women were underrepresented in the group.

Previous studies gave a strong, but not consistent signal, of CPAP lowering cardiovascular risk in OSA patients, especially when the adherence to therapy is satisfactory. The results of our study showed a general decline in inflammation biomarkers, including the new biomarkers and the cardiac biomarker NTproBNP after satisfactory adherence to CPAP therapy in the long term. They suggest a positive effect of the therapy on chronic inflammatory states, thereby possibly reducing cardiovascular risk. There is a possibility that the new inflammatory biomarkers could be used in patient follow up as a cheap and widely available indirect adherence indicator.

## 5. Conclusions

New inflammatory markers such as NLR and FAR are showing positive effects of CPAP therapy on the level of inflammation in patients with severe OSA. CPAP therapy may reduce the risk for CVDs by lowering the level of chronic inflammation.

## Figures and Tables

**Figure 1 jcm-11-06113-f001:**
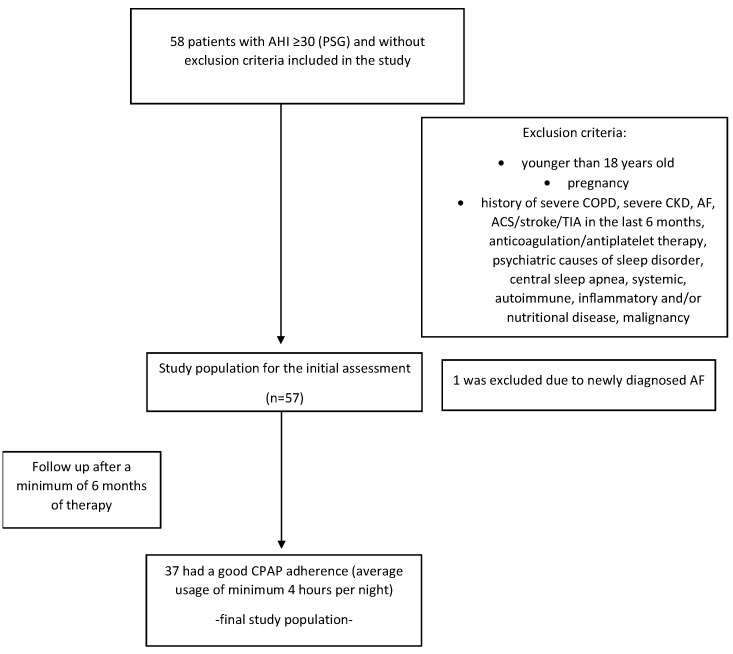
Flow chart of the study population. ACS—acute coronary syndrome, AF—atrial fibrillation, AHI—apnea–hypopnea index, CKD—chronic kidney disease, COPD—chronic obstructive pulmonary disease, CPAP—continuous positive airway pressure, PSG—polysomnography, TIA—transient ischemic attack.

**Figure 2 jcm-11-06113-f002:**
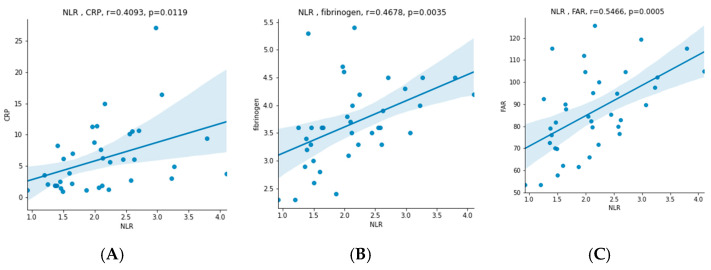
Correlation between neutrophil–lymphocyte ratio (NLR) and (**A**) C-reactive protein (CRP), (**B**) fibrinogen and (**C**) fibrinogen–albumin ratio (FAR).

**Figure 3 jcm-11-06113-f003:**
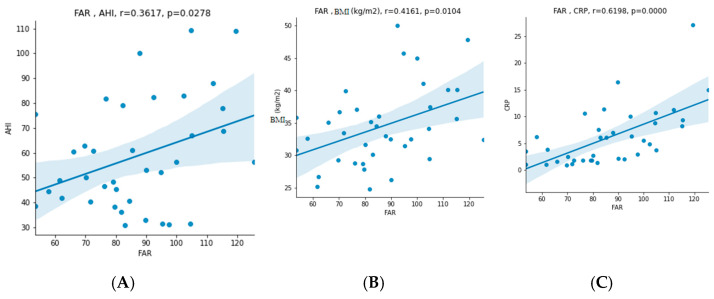
Correlation between fibrinogen–albumin ratio (FAR) and (**A**) apnea–hypopnea index (AHI), (**B**) body mass index (BMI) and (**C**) C-reactive protein (CRP).

**Figure 4 jcm-11-06113-f004:**
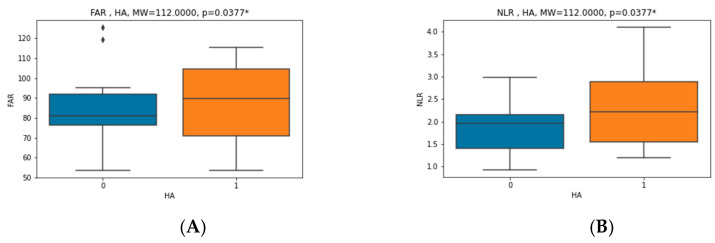
Correlation between (**A**) fibrinogen–albumin ratio (FAR) and hypertension and (**B**) neutrophil–lymphocyte ratio (NLR) and hypertension (HA). *—for meeting statistical significance.

**Figure 5 jcm-11-06113-f005:**
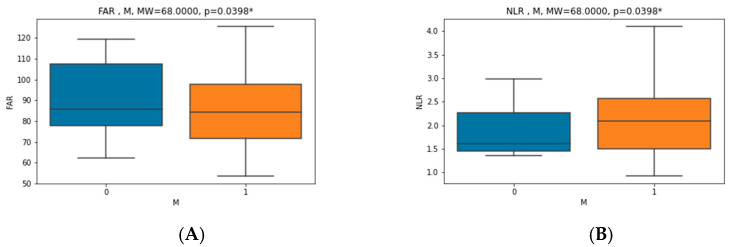
Correlation between (**A**) fibrinogen–albumin ratio (FAR) and gender (M-male) and (**B**) neutrophil–lymphocyte ratio (NLR) and gender (M-male). *—for meeting statistical significance.

**Table 1 jcm-11-06113-t001:** Demographic and clinical characteristics of the study group.

Age (years)	53 ± 10
Male/female gender	29/8
BMI (kg/m^2^)	34.4 ± 6.1
Smoker (number/%)	10/27
Comorbidities	
Hypertension (number/%)	19/51
CVD	1 *
COPD	2
Diabetes mellitus (number/%)	6/16
HbA1c (%)	6.31 ± 1.1
CKD	1 *
eGFR	91.8 ± 13.4
Polysomnographic study results	
AHI (events/h)	58.4 ± 22
CA (events/h)	1.3
OA (events/h)	33.35
Hypopnea (events/h)	11.15
Average apnea duration (sec.)	24.2 ± 6.1
Minimum O^2^ saturation	74.03 ± 11.36
ESS	10.6 ± 5.2
Average time of CPAP therapy per night (min)	322.3 ± 51.3
Duration of CPAP therapy (days)	290.49 ± 56.7

AHI—apnea–hypopnea index, BMI—body mass index, CA—central apnea, CKD—chronic kidney disease, CPAP—continuous positive airway pressure, CVD—cardiovascular disease including acute coronary syndrome, stroke/transient ischemic attack/heart failure, COPD—chronic obstructive pulmonary disease, eGFR—estimated glomerular filtration rate, ESS—Epworth sleepiness scale, HbA1c—glycated hemoglobin, OA—obstructive apnea. * Only one patient had a history of heart failure and mild CKD.

**Table 2 jcm-11-06113-t002:** Clinical laboratory values of the study group.

	Before CPAP Therapy	After CPAP Therapy	*p* Value
Complete blood count parameters			
Leucocyte (×10^9^/L)	7.389 ± 2.034	7.287 ± 2.3	0.6329
Erythrocyte (×10^12^/L)	4.952 ± 0.381	4.904 ± 0.391	0.2482
Hemoglobin (g/L)	150 (141–155)	150 (142–154)	0.401
Hematocrit (L/L)	0.446 ± 0.034	0.444 ± 0.037	0.6261
MCV (fL)	90.189 ± 4.339	90.784 ± 4.626	0.064
MCH (pg)	30 (29–32)	30 (29–32)	0.8672
MCHC (g/L)	336.486 ± 7.633	333.27 ± 8.566	0.0463
RDW (%)	12.9 (12.5–13.6)	13.3 (12.9–13.7)	0.0004
MPV (fL)	8 (7.6–8.3)	8 (7.5–8.5)	0.5502
Thrombocyte (×10^9^/L)	248.459 ± 52.909	239.649 ± 59.237	0.1696
Lymphocyte, %	0.305 ± 0.067	0.331 ± 0.084	0.0009
Lymphocyte (×10^9^/L)	2.24 (1.77–2.51)	2.2 (1.88–2.58)	0.1868
Neutrophil, %	0.6 ± 0.073	0.569 ± 0.09	0.0007
Neutrophil (×10^9^/L)	4.07 (3.14–5.83)	3.83 (2.82–5.11)	0.0214
Biomarkers			
CRP (mg/L)	6.149 ± 5.433	5.254 ± 4.667	0.1698
Albumin (g/L)	42.595 ± 2.713	43.811 ± 3.017	0.0025
NTproBNP (ng/L)	45.27 (28.33–93.81)	35 (30.99–59.88)	0.0427
Fibrinogen (g/L)	3.665 ± 0.752	3.365 ± 0.771	0.0075
Lipid profile			
Total cholesterol (mmol/L)	5.408 ± 0.98	5.052 ± 0.87	0.005
Triglycerides (mmol/L)	1.48 (1.12–2.12)	1.2 (0.92–2.27)	0.338
HDL (mmol/L)	1.2 (1.09–1.47)	1.2 (1.09–1.42)	0.5495
LDL (mmol/L)	3.52 (3.18–4.28)	3.59 (2.92–3.77)	0.0012
OSA-related parameters			
AHI (events/h)	53.2 (40.8–75.5)	4 (2.9–5)	0
ESS	9 (7–15)	4 (3–6)	0
Other			
BMI (kg/m^2^)	34.4 ± 6.1	33.835 ± 5.163	0.1107
Calculated ratios			
PLR	119.586 ± 33.164	110.993 ± 39.775	0.0874
NLR	2.122 ± 0.745	1.888 ± 0.735	0.0008
WMR	920.344 ± 249.522	905.2 ± 280.825	0.579
CAR	0.11 (0.05–0.2)	0.1 (0.02–0.21)	0.1584
FAR	86.445 ± 18.763	77.321 ± 19.133	0.0007

AHI—apnea–hypopnea index, BMI—body mass index, CAR—C-reactive protein–albumin ratio, CPAP—continuous positive airway pressure, CRP—C-reactive protein, ESS—Epworth sleepiness scale, FAR—fibrinogen–albumin ratio, HDL—high-density lipoprotein, LDL—low-density lipoprotein, MCH—mean corpuscular hemoglobin, MCHC—mean corpuscular hemoglobin concentration, MCV—mean corpuscular volume, MPV—mean platelet volume ratio, NLR—neutrophil–lymphocyte ratio, NTproBNP—N-terminal prohormone of brain natriuretic peptide, PLR—platelet–lymphocyte ratio, RDW—red cell distribution width, WMR—white blood cell–mean platelet volume ratio.

## Data Availability

Not applicable.

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
