# Peer review of "Impact of CPAP Therapy on New Inflammation Biomarkers"

_jcm, 2022, doi:10.3390/jcm11206113_

Round 1

Reviewer 1 Report

The paper by Frisic et al. aims to assess the impact of CPAP therapy on systemic inflammation markers in a small cohort of OSA subjects undergoing CPAP treatment.

The paper is globally well-written, and I have only minor suggestions in order to improve it:

- Inflammation plays a role in several pathologies, not only cardiovascular, such as neurodegenerative diseases, since these two entities are strictly related. Moreover, some authors suggested CPAP treatment to improve cognitive function in specific subsets of subjects. Please add a sentence in the discussion on this topic and cite the literature regarding this point.

- Did authors observe any sex-related differences in NLR or other inflammation markers ? Please add a sentence to comment on this point 

- There are some typos and minor grammar errors that authors should correct

Reviewer 2 Report

The paper "Impact of CPAP Therapy on New Inflammation Biomarkers" by Tea Friščić et al. presents the results from a cohort of OSA patients treated with CPAP with the impact of different serum biomarkers. There are some issues that can be improved. 

Page 2, line 64. The outpatient clinic for sleep disorders had an accredited sleep lab? With certified somnologists? PSG is used on hospital sleep laboratories, usually. Did you manually validate the sleep study? Any data about sleep stages, sleep efficiency, arousal index? 

Page 2, line 84. 4 h/night, how many nights/week? Can you comment on results from noncompliant patients? 

Page 4, table 1. Discrepancies in the text and table: AHI 58.4 in the table, 53.2 in the text, ESS 10.6 in the table, 9 in the text. 

Page 6, line 153. number of lymphocytes instead of white blood cell. 

Reviewer 3 Report

The original research entitled “Impact of CPAP Therapy on New Inflammation Biomarkers” is a well written and interesting manuscript with few observations:

Ø  It could be very interesting that baseline and follow-up of the 20 excluded patients were published and compared in a table with the 37 included patients

Ø  In page 9 from lines 274 to 278 the authors discussed their cholesterol results but any comment about statin therapy is reflected

Reviewer 4 Report

First, thank you for reviewing your article, which I found very interesting.

Achieving the most optimal therapy possible will always allow a better quality of life for patients and this will always have a positive influence on them.

Some comments

1 I recommend not to use the same terms, or words, in the title and in the keywords. At the time of the search, the search force is lost. So I suggest you either put new words or not repeat them

2 Comments that a local committee has approved the study. If they can put the number that the committee has given them, it would make it more solid.

3. Comment on the results and discussion of the decrease in NLR, the reduction in neutrophils, and the relative increase in lymphocytes. I would like to see if there are other works that share or not these data using other therapies.

4.- Regarding fibrinogen values. I am left wondering if fibrinogen values ​​cannot be a cause of patient comorbidities and if, how this difference can be made so that this value has significance and is not due to comorbidity.

5. The conclusions seem a bit poor to me. It would be necessary to give them a little more entity

One of the conclusions they draw is that CPAP therapy can reduce the risk of CVD, my doubt is that they have patients in whom the comorbidities they have are the cause of many of these risks. To make this assertion so sure I do not see it clearly.
